# Pediatric diarrhea patients living in urban areas have a higher incidence of *Clostridioides difficile* infection

**Ayodele T. Adesoji**[1], **Osaro Mgbere**[2,3], **Charles Darkoh**[4,5]*

**1** Department of Microbiology, Federal University, Dutsin-Ma, Katsina State, Nigeria, **2** Disease Prevention and Control Division, Houston Health Department, Houston, Texas, United States of America, **3** Institute of Community Health, Texas Medical Center, University of Houston College of Pharmacy, Houston, Texas, United States of America, **4** Department of Epidemiology, Human Genetics, and Environmental Sciences, School of Public Health, Center for Infectious Diseases, University of Texas Health Science Center, Houston, TX, United States of America, **5** MD Anderson Cancer Center UTHealth Graduate School of Biomedical Sciences, Microbiology and Infectious Diseases Program, Houston, Texas, United States of America

\* charles.darkoh@uth.tmc.edu

**Data Availability Statement:** The datasets generated and/or analyzed during the current study are not publicly available but may be made available upon request from the corresponding author or from the UTHealth Committee for the

## Abstract

*Clostridioides difficile* infection (CDI) is a major cause of antibiotic-associated diarrhea and an unappreciated contributor to child mortality in low- and middle-income countries where the diagnosis may be difficult. There is little information about the prevalence of CDI among infants, children, and adolescents in Africa. Using a cross-sectional design, seventy-six samples were collected from pediatric patients presenting with diarrhea, including infants ($\leq$ 2 years old), children (2–12 years), and adolescents (13 $\leq$17 years) from three hospitals between January and December 2019. Demographic data, medical history, and prior antibiotic use were recorded. Toxigenic culture and PCR were used to detect and validate the presence of *C. difficile* in the samples. Data obtained were analyzed using descriptive and inferential statistics. A total of 29 (38.7%), 39 (52.0%) and 7 (9.3%) samples were from infants, children, and adolescents, respectively. The average age of the patients was 4.4 years. Of these samples, 31 (41%) were positive for *C. difficile* by culture and were verified by PCR amplification of *C. difficile-specific* genes (*tcdA* and *tcdB*). The most positive cases were children (53.3%) and infants (40.0%) with the majority of them residing in urban areas. Forty-nine (66.2%) of the patients had no known antibiotics exposure, whereas 29.0% and 29.7% reported the use of over-the-counter antibiotics at 14 and 90 days, prior to the hospital visit, respectively. CDI is relatively common among children with diarrhea in Northern Nigeria. Therefore, for effective management and treatment, more attention should be given to testing for *C. difficile* as one of the causative agents of diarrhea.

## Introduction

*Clostridioides difficile* is a strict anaerobic Gram-positive bacterium that colonizes up to 15% of healthy people [1]. It is a major cause of antibiotic-associated diarrhea worldwide, whereby antibiotic treatment reduces the abundance of competing gut microbiota, allowing *C. difficile* to proliferate and cause mild to severe diarrhea [2–4].

Protection of Human Subjects at cphs@uth.tmc. edu.

**Funding:** This study was supported in part by the National Institutes of Allergy and Infectious Diseases (NIH/NIAID) grants R01AI116914 [CD] and R01AI150685 [CD]. The funders had no role in study design, data collection, and analysis, decision to publish, or preparation of the manuscript.

**Competing interests:** The authors have declared that no competing interests exist.

The toxins (enterotoxin A and cytotoxin B) released during *C. difficile* infection (CDI) cause tissue damage [5]. These two toxins, encoded by the *tcdA* and *tcdB*, are located within a 19.6 kbp pathogenicity locus. Older age, antibiotic usage, gastric acid-suppressing drugs, inflammatory bowel disease, gastrointestinal surgery, use of naso, gastric tubes, neoplastic disease, immunodeficiency, and comorbidities are risk factors for CDI [6].

The United States Center for Disease Control and Prevention (CDC) declared *C. difficile* as one of the five 'urgent health threats in 2019, because of the significant risk associated with antibiotic use [7]. Despite this urgency, the prevalence of CDI in low and middle-income countries is underestimated. In developing nations where self-medication occurs frequently, a higher incidence of CDI may be expected. Outbreaks of CDI in developed countries have increased vigilance in these countries but little is known about CDI in Nigeria [8]. Adegboyega [9] and Doughari, Kachalla and Jaafaru, [10] found *C. difficile* spores and the bacterium, respectively, from samples obtained from hospital environments. Oguike and Emeruwa [11] isolated *C. difficile* from 156 (48.8%) out of 320 stool samples collected from infants under the age of 5 years and confirmed the occurrence of cytotoxin production from 25 (14.8%) of the isolates. Onwueme et al., [12] reported that among 71 HIV outpatients at one hospital, 10 (14.1%) and 61 (85.9%) of *C. difficile* isolates were toxin-positive and toxin-negative, respectively.

Normally, CDI testing in developing countries is not routine owing to the lack of resources for diagnosis and culture facilities for obligate anaerobes [13]. This is unfortunate because the majority of global mortality among children in the first 5 years of their life is due to diarrhea, of which a potentially large proportion could involve CDI [14, 15]. Limited testing also translates into a low index of clinical suspicion that exacerbates the lack of effort toward diagnostic screening [12, 16] resulting in misdiagnosis and incorrect treatment [17]. The lack of information about CDI is also problematic because most African countries are popular tourist destinations and may serve as incubators of CDI [18].

In most African countries, antimicrobials are easily accessed over-the-counter (OTC) [17, 19, 20]. There is also a high incidence of community- and nosocomial-associated diarrhea among children [20, 21], but few data are available on the incidence of CDI. Consequently, we initiated a survey to estimate the occurrence of CDI among pediatric diarrheic outpatients (0–17 years) in Katsina State, Nigeria.

## Materials and methods

### Study population and sample collection

Using a cross-sectional design, a total of 76 participants from three hospitals in Katsina State including (Malam Mande General Hospital in Dutsin-Ma) and Turai Yaradua Maternity and Children hospital and Government House Clinic in Katsina metropolis) were sampled for this study. The Katsina metropolis was designated as an urban area, while Dutsin-Ma was designated as a rural area. All the hospitals are under the management of the Katsina State Hospital Management Board.

Only diarrhea-presenting out-patients who were reporting to these hospitals between January and December 2019, and between the age of 0 and 17 years old (infant <2 years, children 2–12 years, adolescent 13–17 years) were eligible for inclusion in the study. The case definition was based on evaluation by attending physicians, where the severity of illness was indicated by whether a child was admitted to the hospital (inpatient) or treated as an outpatient.

As a result of the non-availability of information on diarrhea frequency or gravity of the symptoms available, disease severity was determined by the treatment setting of the patients [22]. Accordingly, diarrhea that required hospitalization was classified as severe, those that

resolved within a few hours were considered mild, while the ones that resolved within a day were classified as moderate.

Recruitment of the patients was integrated with the services of the clinic and hospital ward under the supervision of the attending physicians who referred diarrhea patients to the microbiology laboratory. Following consent, questionnaires were administered to the patients that captured demographic data, previous antibiotics use, and medical history including hospitalization, and then stool samples were collected and stored at -20˚C. The samples were subsequently shipped on ice to the University of Texas Health Science Center, School of Public Health Center for Infectious Diseases, Houston, Texas, USA for analysis.

## Detection of *C. difficile* in stool samples by culture

To detect *C. difficile*, the stools were tested by toxigenic culture [23], followed by PCR of the isolates, as described previously [18]. Anaerobic condition was maintained in a Bactron 600 anaerobic chamber (Sheldon Manufacturing, Cornelius, OR, USA) using 5% $CO_2$, 10% $H_2$, and 85% $N_2$.

## PCR analysis of bacterial pellets

DNA from the isolates was extracted using the GenElute Bacterial Genomic DNA Kit (Sigma-Aldrich, St Louis, MO, USA), according to the manufacturer's instructions. The concentration of the extracted DNA was estimated using a NanoDrop (ThermoScientific Wilmington, DE, USA). Primers specific for toxins A and B were used to assess the presence of these genes while a 16S ribosomal marker (16S rRNA) served as a positive control for the PCR reaction [15, 24–29]. The primer sequences used were as follows: TcdA2 (F-5′AGATTCCTATATTTACATGACAATAT3′, R-5′GTATCAGGCATAAAGTAATATACTTT3′); TNC (F-5′GAGCACAAAGGGTATTGCTC–TACTGGC3′, R-5′CCAGACACAGCTAATCTTATTTGCACCT3′); 16S rRNA (F-ACACGGTC–CAAACTCCTACG, R-5′AGGCGAGTTTCAGCCTACAA3′). OneTaq Quick-Load 2X Master Mix (New England Biolabs, IPSwich, MA, USA) was used for the PCR amplification with an initial denaturation temperature of 94˚C for 30 s, followed by 36 cycles of 94˚C for 30s; 55˚C for 30s; 68˚C for 30s; and a final extension of 68˚C for 5 min. Gel electrophoresis (1.0%) followed by staining with ethidium bromide and UV exposure was used to assess the results.

## Toxin detection

Cdifftox activity assay was used to detect toxin activity using 48 h culture supernatant from the isolates [30] while C. *difficile* TOX A/B II ELISA test (TechLab, Blacksburg, VA, USA) was used to detect toxin production.

For the ELISA test, 200 µl of the culture supernatant was treated according to the manufacturer's instructions.

## Statistical analysis

Data obtained from the survey were subjected to descriptive statistics using frequency runs to describe the proportional distribution of the study sample by sex, age educational status, and area of residence. To determine the associations between the history of antibiotic use, the severity of diarrhea, and the presence of toxigenic *C. difficile* in the stool samples of the patients, we applied a chi-square inferential test of independence. Furthermore, to examine the relationships between the length of diarrhea and hospitalization and the age of children with CDI, we carried out bivariate curve fitting models and density contour mappings to determine the potential patterns (clusters) within these measures. Density ellipse probability

was applied to evaluate the mass of points of the measures at $p = 0.25$ and $p = 0.50$. Additionally, we used Pearson's correlation coefficient to determine the linear correlation between each of the measures and the age of diarrheal patients. Based on the sample data, we build a partition-based model using the participants' area of residence, age category, and sex to determine the probability of obtaining a positive CDI test result. All statistical tests conducted were 2-tailed, and a probability of ≤0.05 was used as the threshold for declaring statistical significance. Data management and statistical analyses were conducted using SAS JMP Statistical Discovery Software version 14.3 (SAS Institute, Cary, North Carolina, USA).

## Ethics statement

Approval to conduct this study was obtained from the Ethical Review Committee of the Ministry of Health in Katsina State, Nigeria (MOH/ADM/SUB/1152/258) after reviewing the study protocol that included the consent procedures. Written consent from responsible guardians and 'assent' for those >7 years old was obtained before sample collection. Guardians of participants were informed of the purpose of the study and the fact that participation in the survey was anonymous, consensual, and voluntary.

## Results

Seventy-six stool samples were collected from patients presenting with diarrhea: 29 (38.7%), 39 (52.0%), and 7 (9.3%) infants, children, and adolescents, respectively (Table 1). There were more male (63.2%) than female (36.8%) patients.

For all samples, 31 (41%) grew *C. difficile* colonies while 45 (59%) were negative. All putative *C. difficile* colonies were PCR-positive for *tcdA* and *tcdB* genes, and toxin positive by ELISA. None of the isolates that tested negative by PCR were positive for toxin production. Among the confirmed *C. difficile* patients, 11 (35%) and 20 (65%) were female and male,

**Table 1. Demographic characteristics of the pediatric patients with diarrhea.**

| Characteristic | N | % |
|---|---|---|
| **Sex** | | |
| *Female* | 28 | 36.8 |
| *Male* | 48 | 63.2 |
| **Age Category (years)** | | |
| *Infant (< 2)* | 29 | 38.7 |
| *Children (2–12)* | 39 | 52.0 |
| *Adolescent (13–17)* | 7 | 9.3 |
| Mean ± SEM (4.4 ± 0.51) | | |
| **Educational Status** | | |
| *No Schooling* | 7 | 9.2 |
| *Primary School* | 9 | 11.8 |
| *Secondary School* | 8 | 10.5 |
| *Unknown* | 52 | 68.4 |
| **Area of Residence** | | |
| *Urban* | 44 | 57.9 |
| *Rural* | 32 | 42.1 |

Note: Within characteristic, n may not add up to the total number of records due to missing response; Percentages may not add up to 100 due to rounding.

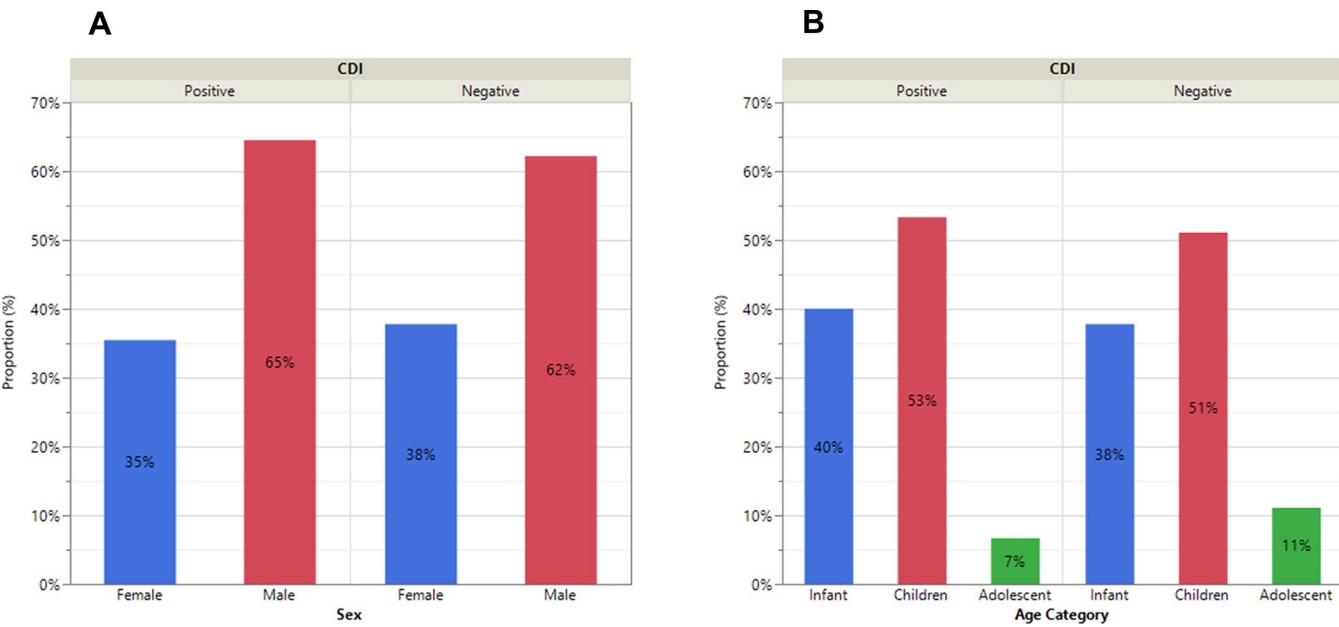

**Fig 1.** Distribution of the presence of *C. difficile* in the stools of pediatric patients with diarrhea by sex (A) and age (B).

respectively (Fig 1A). The distributions across infants, children, and adolescents were 40%, 53%, and 7%, respectively (Fig 1B).

The majority of the parents of the patients (66.2%) did not have any knowledge about antibiotics, whereas 29.0% and 29.7% reported OTC antibiotic use at 14 and 90 days respectively. About 13.2% and 9.2%, respectively, reported using metronidazole, and other types of antibiotics (tetracycline, ampicillin, ampiclox, cloxacillin, ciprofloxacin, gentamicin, septrin, cefuroxime, erythromycin, doxycycline, penicillin, etc.). There was a significant association between knowledge of antibiotics ($X^2 = 3.74$, $p = 0.05$) and use of antibiotics in the last 14 days ($X^2 = 4.56$, $p = 0.05$), and the occurrence of CDI (Table 2). Of those who have knowledge of antibiotics (n = 25, 33.8%), 18.9% of them tested positive for CDI, while 17.7% (n = 11) of the patients with positive CDI indicated having taken antibiotics in the last 14 days ($p = 0.05$) prior to sample collection.

Among the patients, 19.7%, 66.2%, and 14.1% reported mild, moderate, and severe diarrhea cases, respectively. Also, 55.8% (n = 24) and 44.2% (n = 19) reported suffering from diarrhea in the past 14 and 30 or more days prior to their hospital visit, respectively (Table 2). Some patients (32.9%) exhibited symptoms of vomiting prior to their visit to the clinic. The majority of the patients (68.1%) reported no prior hospitalization due to diarrhea.

Fig 2A and 2B demonstrate the relationships between length of diarrhea, hospitalization, and age of the pediatric patients with the contours displaying the regions with various densities. The contours with red color and gradients denote regions of high density and fade to regions with medium densities colored light green, and subsequently, to regions of low densities colored purple with gradients. The inner and outer density ellipses represent regions where 25% and 50% of the cases, respectively occurred in relation to the measures. The average length of diarrhea among the patients was 5.75±1.25 days at an average age of 4.36±0.52 years, while the average length of hospitalization due to diarrhea was 7.06±3.57 days (1 week) at an average age of 3.36±0.91 years. Although the length of diarrhea and hospital stay were positively and significantly correlated ($r = 0.6870$, $p < 0.01$), they were both negatively correlated

**Table 2. History of antibiotics used by the patients, presence of toxigenic *C. difficile*, and disease severity among pediatric patients with diarrhea.**

| Measure | N (%) | CDI | |
|---|---|---|---|
| | | Positive | Negative |
| | | n (%) | n (%) |
| **Parent's Knowledge of Antibiotics** | | | |
| *No* | 49 (66.2) | 16 (21.6) | 33 (44.6) |
| *Yes* | 25 (33.8) | 14 (18.9) | 11 (14.9) |
| Test Statistics: $\chi 2$ value, *P-value* | | 3.743, 0.0530* | |
| **Use Antibiotics in the last 14 days** | | | |
| *No* | 44 (71.0) | 14 (22.6) | 30 (48.4) |
| *Yes* | 18 (29.0) | 11 (17.7) | 7 (11.3) |
| Test Statistics: $\chi 2$ value, *P-value* | | 4.555, 0.0328* | |
| **Type of Antibiotic Used in the last 14 days** | | | |
| *Metronidazole/Flaggy* | 10 (13.2) | 7 (9.2) | 3 (4.0) |
| *Other types of Antibiotics* | 7 (9.2) | 4 (5.3) | 3 (4.0) |
| *Unknown* | 59 (77.6) | 20 (26.3) | 39 (51.3) |
| Test Statistics: $\chi 2$ value, *P-value* | | 5.468, 0.0650 | |
| **Used Antibiotics in the Last 90 days (3 Months)** | | | |
| *No* | 52 (70.3) | 20 (27.0) | 32 (43.2) |
| *Yes* | 22 (29.7) | 11 (14.9) | 11 (14.9) |
| Test Statistics: $\chi 2$ value, *P-value* | | 0.846, 0.3578 | |
| **Last occurrence of diarrhea prior to clinic visit** | | | |
| *Less than 14 days* | 24 (55.8) | 15 (34.9) | 9 (20.9) |
| *Last 1 or more months* | 19 (44.2) | 7 (16.3) | 12 (27.9) |
| Test Statistics: $\chi 2$ value, *P-value* | | 2.794; 0.0946 | |
| **Ailment Experienced prior to Clinic visit** | | | |
| *Vomiting* | 25 (32.9) | 11 (14.5) | 14 (18.4) |
| *Non-vomiting* | 30 (39.5) | 10 (13.2) | 20 (26.3) |
| *Unknown* | 21 (27.6) | 10 (13.2) | 11 (14.5) |
| Test Statistics: $\chi 2$ value, *P-value* | | 1.203; 0.5480 | |
| **Severity of Diarrhea** | | | |
| *Mild* | 14 (19.7) | 7 (9.9) | 7 (9.9) |
| *Moderate* | 47 (66.2) | 18 (25.4) | 29 (40.9) |
| *Severe* | 10 (14.1) | 3 (4.2) | 7 (9.9) |
| Test Statistics: $\chi 2$ value, *P-value* | | 1.052; 0.5908 | |
| **Hospitalization due to Diarrhea** | | | |
| *No* | 49 (68.1) | 20 (27.8) | 29 (40.3) |
| *Yes* | 23 (31.9) | 9 (12.5) | 14 (19.4) |
| Test Statistics: $\chi 2$ value, *P-value* | | 0.018; 0.8918 | |

Note: Within measure percentages may not add up to 100 exactly due to rounding; within characteristic, "n" may not add up to the total number of records due to missing response.

Significance Level

* = p<0.05; ns = not significant (p>0.05).

with the patients' age ($r$ = -0.1946 vs. -0.1336) (*Results not presented*). Consequently, younger children were more likely to have longer diarrheal periods and hospitalization than older ones.

Fig 3 shows the probability of obtaining a positive CDI test by selected demographic characteristics. The majority of the children residing in the urban areas had more positive CDI

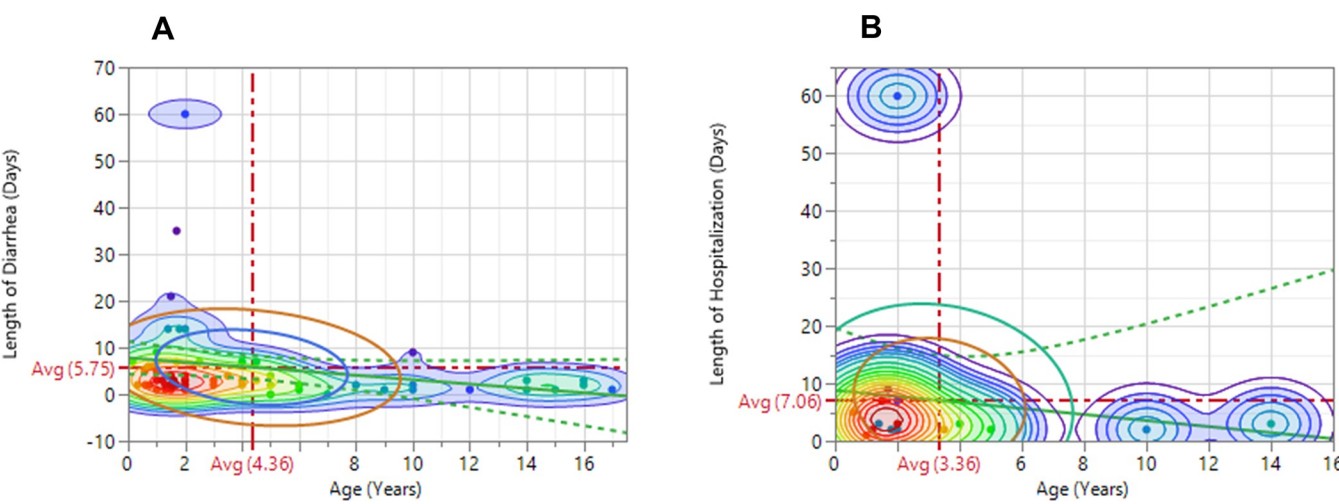

**Fig 2.** Relationship between the length of diarrhea and hospitalization (A) and age of pediatric patients (B).

compared to those living in the rural areas. For instance, the probability of obtaining a positive CDI among female and male children, of age 2–12 years in urban areas were 0.543 and 0.629 compared to 0.173 and 0.216 for those in the rural areas, respectively.

## Discussion

CDI has been underestimated by clinicians and public health workers in Africa despite a well-documented risk of diarrhea and self-medication with antibiotics [31]. Due to a lack of culture facilities for obligate anaerobes and resources for diagnostic testing, routine tests for CDI are rarely carried out in developing countries, and diarrhea is often treated symptomatically, resulting in mistreatment, misdiagnosis, and likely underrating *C. difficile* contribution to diarrhea [13]. Among the 76 pediatric stool samples tested in this study, 31 (41%) were positive for *C. difficile*. This is within the prevalence rate reported by Emeruwa and Oguike [32], but slightly higher than what Plant-Paris et al., [31] reported among children less than 5 years in Kenya. Our data also showed a higher (53.33%) CDI proportion in children between 2–12 years old compared to 3–15 years old patients (30%) reported by Mutai et al., [13]. Whereas higher observation of CDI among pediatric diarrhea patients living in the urban area compared to rural dwellers in our study is contrary to what Oguike and Emeruwa [11] reported in

| Leaf Label | CDI | | | |
|---|---|---|---|---|
| | + | Plot | - | Plot |
| Area of Residence (Rural) & Children (2-12 yrs) & Sex (Female) | 0.1726 | | 0.8274 | |
| Area of Residence (Rural) & Children (2-12 yrs) & Sex (Male) | 0.2164 | | 0.7836 | |
| Area of Residence (Rural) & Adolescent (13-17 yrs), Infant (<2 yrs) | 0.5251 | | 0.4749 | |
| Area of Residence (Urban) & Adolescent (13-17 yrs), Infant (<2 yrs) & Sex (Female) | 0.2400 | | 0.7600 | |
| Area of Residence (Urban) & Adolescent (13-17 yrs), Infant (<2 yrs) & Sex (Male) | 0.3385 | | 0.6615 | |
| Area of Residence (Urban) & Children (2-12 yrs) & Sex (Female) | 0.5431 | | 0.4569 | |
| Area of Residence (Urban) & Children (2-12 yrs) & Sex (Male) | 0.6288 | | 0.3712 | |

**Fig 3. Response probability of toxigenic CDI among pediatric patients with diarrhea.**

their study. The authors reported higher (48.3–52.5%) CDI among infants living in rural areas than their urban counterparts (42–47.8%). However, in a study conducted recently in rural Ontario, Canada by Babey et al., [33], the authors noticed similarities between CDI incidence in their study compared to urban-based estimates.

We confirmed toxins A and B production in the isolates, suggesting that infants and children can be reservoirs of clinically relevant strains of *C. difficile*. The study also showed that younger children can have longer diarrheal periods and hospitalization than older children. This may be related to the immune system since infants and children still have compromised immune systems compared to adolescents [34]. As a result, diarrhea could be more quickly resolved in adolescents than in infants and children. Also, it has been reported that the prevalence of *C. difficile* colonization in neonates ranges from 2% to 50% with colonization often occurring within the first week of life [35]. In contrast to adults where the rate of colonization in community and hospital settings is 3% and 20%, respectively.

Our study has some limitations that warrant cautious interpretation of the findings. First, the small sample size of participants stemmed from the fact that only diarrheal patients who reported to the hospitals were screened by the attending physicians. Consequently, although the sample size met the statistical threshold and provided adequate statistical power, the non-random sampling technique applied placed some constraints on the generalizability of our findings. This is because some patients who had diarrhea may not have reported to the hospitals. Second, the study population was from one major race/ethnicity in northern Nigeria, making it impossible to determine racial/ethnic disparities in the occurrence and severity of CDI. Third, the presence of other diarrhea-causing pathogens was not tested in this study and will be the subject of our ongoing research in this population. Finally, the survey was self-reported, and thus, some participants may have exhibited social desirability and recall biases in their responses. Despite, these, we believe that social desirability was not, however, associated with the objective measures in our study.

In conclusion, attention should be given to testing for *C. difficile* as one of the causes of diarrhea during diagnosis rather than simply administrating antibiotics empirically. Testing can also promote the development and implementation of strategies that regulate antimicrobials known to induce community and hospital-acquired CDI. As this is the first report of CDI in the northern part of Nigeria, future work will include examining the diversity of strains as well as other phenotypic and virulence-associated characteristics.

## Acknowledgments

The authors would like to thank Malam Mande General Hospital, Turai Yaradua Maternity and Children Hospital, and Government House Clinic managements for facilitating the recruitment of patients for this study.

## Author Contributions

**Conceptualization:** Ayodele T. Adesoji, Osaro Mgbere, Charles Darkoh.

**Data curation:** Ayodele T. Adesoji, Osaro Mgbere.

**Formal analysis:** Ayodele T. Adesoji, Osaro Mgbere, Charles Darkoh.

**Funding acquisition:** Charles Darkoh.

**Investigation:** Ayodele T. Adesoji.

**Methodology:** Ayodele T. Adesoji, Osaro Mgbere.

**Project administration:** Ayodele T. Adesoji.

**Supervision:** Osaro Mgbere, Charles Darkoh.

**Writing – original draft:** Ayodele T. Adesoji, Charles Darkoh.

**Writing – review & editing:** Ayodele T. Adesoji, Osaro Mgbere, Charles Darkoh.

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
