## [Decision Letter · Decision Letter 0]

24 Aug 2022

PGPH-D-22-00619

Pediatric Diarrhea Patients Living in Urban Areas Have Higher Incidence of Clostridioides difficile Infection

Dear Dr. Darkoh,

Thank you for submitting your manuscript to PLOS Global Public Health. After careful consideration, we feel that it has merit but does not fully meet PLOS Global Public Health’s publication criteria as it currently stands. Therefore, we invite you to submit a revised version of the manuscript that addresses the points raised during the review process.

Please find my detailed comments (as an attachment) added along with the reviewers' comments. Overall, this is a very valuable and important study from Nigeria. You will need to address the comments from the two reviewers as well as those from the academic editor.However, a significant modification of the manuscript is needed to make this eligible for publication. 

We look forward to receiving your revised manuscript.

Kind regards,

Sindhu Kulandaipalayam Natarajan, MD, DTM&H

Academic Editor

Journal Requirements:

1. Please provide additional details regarding participant consent. In the ethics statement in the Methods and online submission information, please ensure that you have specified whether consent was written or verbal/oral. If consent was verbal/oral, please specify: 1) whether the ethics committee approved the verbal/oral consent procedure, 2) why written consent could not be obtained, and 3) how verbal/oral consent was recorded. If your study included minors, please state whether you obtained consent from parents or guardians in these cases. If the need for consent was waived by the ethics committee, please include this information.

2. Please send a completed 'Competing Interests' statement, including any COIs declared by your co-authors. If you have no competing interests to declare, please state "The authors have declared that no competing interests exist". Otherwise please declare all competing interests beginning with the statement "I have read the journal's policy and the authors of this manuscript have the following competing interests:"

3. Please amend your detailed Financial Disclosure statement. This is published with the article. It must therefore be completed in full sentences and contain the exact wording you wish to be published.

4. Please provide separate figure files in .tif or .eps format only and remove any figures embedded in your manuscript file. Please also ensure that all files are under our size limit of 10MB.

5. Please provide a complete Data Availability Statement in the submission form, ensuring you include all necessary access information or a reason for why you are unable to make your data freely accessible. If your research concerns only data provided within your submission, please write "All data are in the manuscript and/or supporting information files" as your Data Availability Statement.

Additional Editor Comments (if provided):

Reviewers' comments:

Reviewer's Responses to Questions

**Comments to the Author**

1. Does this manuscript meet PLOS Global Public Health’s publication criteria? Is the manuscript technically sound, and do the data support the conclusions? The manuscript must describe methodologically and ethically rigorous research with conclusions that are appropriately drawn based on the data presented.

Reviewer #1: Yes

Reviewer #2: Yes

2. Has the statistical analysis been performed appropriately and rigorously?

Reviewer #1: Yes

Reviewer #2: I don't know

3. Have the authors made all data underlying the findings in their manuscript fully available (please refer to the Data Availability Statement at the start of the manuscript PDF file)?

Reviewer #1: Yes

Reviewer #2: No

4. Is the manuscript presented in an intelligible fashion and written in standard English?

Reviewer #1: No

Reviewer #2: Yes

5. Review Comments to the Author

Reviewer #1: 1. In the abstract please mention your study design, and a sentence about the statistical methods you used.

2. Study Population: Please include your study design.

3. How did you determine this sample size? Please describe your sampling procedure. And justify whether 76 sample size provide the adequate level of power for this study.

4. Since you have considered three different pediatric groups as your study participant, so how did you adjust for their individual affects on CDI?

5. It is important to know that whether your sample size is adequate enough for performing the statistical test and showing sufficient statistical power to answer your research objectives.

6. In line 151, you have mentioned that frequency distribution of study sample are shown by race/ ethnicity which didn't display in the results section.

7. The level of statistical significance was considered as <=0.05 in this study. According to this cut off-point, the association between knowledge of antibiotics and CDI occurrence are not statistically significant at 5% level of significance.

8. In line 197-200, please describe a little more about the patterns exhibits in the figure 2a, 2b.

9. In figure2, please specify the color legends.

10. Please work on the grammar and sentence structure of the manuscript.

Reviewer #2: It is a well-written research paper which highlights an important issue of Clostridium difficile infection (CDI) among children in low and middle-income countries. The authors should highlight the reasons for the non-diagnosis of CDI in Nigeria from the health system perspective also. Is it a shortage of trained staff or sophisticated equipment which are expensive or so on? This should be included in the discussion section. Secondly, as multiple diagnostic tests are mentioned in the paper, are there any standard treatment guidelines (STGs) for the diagnosis of CDI in Nigeria or other LMICs? if yes, details of diagnostics required for the diagnosis of CDI in a real-world scenario should be added in the introduction of the article. Thirdly, as mentioned in the limitations why the presence of other diarrhoea-causing pathogens was not tested in this study? Does it mean that it can be a non-CDI infection but due to lack of testing classified as a CDI? Can be a multiple pathogen infection? If yes, how authors decided that it's a CDI and no other pathogen is predominant? Lastly, the limitations section is written very briefly, more details can be added to it.

6. PLOS authors have the option to publish the peer review history of their article (what does this mean?). If published, this will include your full peer review and any attached files.

**Do you want your identity to be public for this peer review?** For information about this choice, including consent withdrawal, please see our Privacy Policy.

Reviewer #1: **Yes: **Sudipta Das Gupta

Reviewer #2: **Yes: **Maninder Pal Singh

---

## [Decision Letter · Decision Letter 1]

21 Dec 2022

Pediatric Diarrhea Patients Living in Urban Areas Have Higher Incidence of Clostridioides difficile Infection

PGPH-D-22-00619R1

Dear Dr Darkoh,

We are pleased to inform you that your manuscript 'Pediatric Diarrhea Patients Living in Urban Areas Have Higher Incidence of Clostridioides difficile Infection' has been provisionally accepted for publication in PLOS Global Public Health.

Best regards,

Sindhu Kulandaipalayam Natarajan, MD, DTM&H

Academic Editor

Reviewer Comments (if any, and for reference):

Reviewer's Responses to Questions

**Comments to the Author**

1. If the authors have adequately addressed your comments raised in a previous round of review and you feel that this manuscript is now acceptable for publication, you may indicate that here to bypass the “Comments to the Author” section, enter your conflict of interest statement in the “Confidential to Editor” section, and submit your "Accept" recommendation.

Reviewer #1: All comments have been addressed

2. Does this manuscript meet PLOS Global Public Health’s publication criteria? Is the manuscript technically sound, and do the data support the conclusions? The manuscript must describe methodologically and ethically rigorous research with conclusions that are appropriately drawn based on the data presented.

Reviewer #1: Yes

3. Has the statistical analysis been performed appropriately and rigorously?

Reviewer #1: Yes

4. Have the authors made all data underlying the findings in their manuscript fully available (please refer to the Data Availability Statement at the start of the manuscript PDF file)?

Reviewer #1: Yes

5. Is the manuscript presented in an intelligible fashion and written in standard English?

Reviewer #1: Yes
